# Sandbagging in a Simple Survival Bandit Problem

**Joel Dyer**[*]
University of Oxford

**Daniel Jarne Ornia**
University of Oxford

**Nicholas Bishop**
University of Oxford

**Anisoara Calinescu**
University of Oxford

**Michael Wooldridge**
University of Oxford

## Abstract

Evaluating the safety of frontier AI systems is an increasingly important concern, helping to measure the capabilities of such models and identify risks before deployment. However, it has been recognised that if AI agents are aware that they are being evaluated, such agents may deliberately hide dangerous capabilities or intentionally demonstrate suboptimal performance in safety-related tasks in order to be released and to avoid being deactivated or retrained. Such strategic deception – often known as "sandbagging" – threatens to undermine the integrity of safety evaluations. For this reason, it is of value to identify methods that enable us to distinguish behavioural patterns that demonstrate a true lack of capability from behavioural patterns that are consistent with sandbagging. In this paper, we develop a simple model of strategic deception in sequential decision-making tasks, inspired by the recently developed survival bandit framework. We demonstrate theoretically that this problem induces sandbagging behaviour in optimal rational agents, and construct a statistical test to distinguish between sandbagging and incompetence from sequences of test scores. In simulation experiments, we investigate the reliability of this test in allowing us to distinguish between such behaviours in bandit models. This work aims to establish a potential avenue for developing robust statistical procedures for use in the science of frontier model evaluations.

## 1 Introduction

As frontier AI models become more and more capable, concerns arise over the potential of such models to develop and carry out increasingly complex plans for pursuing dangerous or misaligned goals they might have. For this reason, pre-deployment procedures for evaluating the safety of frontier AI models – which aim to test the dangerous capabilities of such models and, upon conclusion of these evaluations, make decisions about whether they are to be deployed, retrained, or withheld from broader use, for example – are under active development [see, e.g., AI Security Institute, 2024].

However, it has been recognised that if AI systems become aware of the evaluation mechanism during safety tests, such systems might engage in a form of strategic deception often termed "sandbagging", in which the model underplays their abilities in dangerous tasks to avoid being classed as exceedingly dangerous and consequently decommissioned or retrained [van der Weij et al., 2024]. Such deception threatens to undermine the trustworthiness of safety evaluations, and increases the risk of potentially catastrophic consequences if unsafe models are inadvertently released. To combat this, it is of value to develop methods to predict and detect sandbagging behaviours in evaluation settings.

---

[*]`joel.dyer@cs.ox.ac.uk`

39th Conference on Neural Information Processing Systems (NeurIPS 2025) Workshop: Reliable ML from Unreliable Data.

**Contribution**  In this paper, we formulate an evaluation process for a frontier AI system consisting of a sequence of tasks that measure its dangerous capabilities as a sequential decision-making problem. In particular, we propose a modified version of the survival bandit problem [Riou et al., 2022], in which we suppose that a dangerous and misaligned agent pursues its objectives during an evaluation procedure, where the evaluation determines that the agent is too dangerous if its cumulative performance during the evaluation period exceeds some safety threshold (see Section 3). We study this model theoretically, establishing theoretical conditions under which a rational actor acting optimally in this setting will engage in sandbagging behaviour (see Section 4), and verify this behaviour in numerical experiments (see Section 5). On the basis of this model, we propose and study a simple statistical test to distinguish sandbagging from a genuine lack of dangerous capability, using observations of the reward sequence alone and without assuming access to the AI model itself (see Section 6). We argue that, since advanced reasoning agents may explicitly model their environment according to the model we formulate, this framework offers a potentially valuable lens for interpreting agent behavior and for designing statistical tools to detect deception in safety evaluations.

## 2  Background

**Sandbagging in AI capability evaluations**  Recent work shows that frontier AI models can "sandbag" – strategically underperform on capability evaluations – thereby obscuring dangerous competencies while maintaining strong performance elsewhere, and undermining the trustworthiness of safety assessments [van der Weij et al., 2024]. This possibility has motivated the development of techniques to detect sandbagging. For example, Tice et al. [2024] recognise that adding small amounts of Gaussian noise to the networks weights of large language models that are prompted or fine-tuned to sandbag can improve their performance during capability evaluations, indicating that such so-called "noise injections" applied to model weights can be useful for detecting sandbagging. Our work differs in that we study an explicit theoretical model that induces sandbagging in optimal rational agents, and use our model to construct a sandbagging detection method on the basis of the model behaviour alone and without assuming access to model weights. Such "black-box" methods are valuable in situations in which access to model weights is not given, for example language models that are accessible only via APIs.

**Survival bandits**  The survival bandit (S-MAB) problem [Riou et al., 2022] is a variant of the classic multi-armed bandit (MAB) problem [Lattimore and Szepesvári, 2020], in which the decision-making process is halted if the cumulative reward falls below a predefined threshold. In contrast to the classical MAB problem, the S-MAB problem emphasises the trade-off between performance and survival under constraints on the resource available to the decision-making agent. The S-MAB problem has more recently been considered as a framework for modelling the emergence of misalignment in principal-agent problems as a result of mismatches in resource constraints, with implications for the deployment of aligned AI agents [Ornia et al., 2025]. We deviate from these works by formulating and studying a modification to the basic S-MAB problem that more faithfully reflects the nature of the safety evaluation process for frontier AI models.

## 3  Model definition

We model the evaluation process as a sequential decision-making task in which the agent being evaluated completes a number of tasks in sequence that assess its dangerous capabilities. Let $\mathcal{Y}$ be a finite set of outcomes and $\mathcal{A}$ a finite set of actions that the agent can take in each round of the evaluation process. Each $a \in \mathcal{A}$ is associated with a distribution $p_a$ over $\mathcal{Y}$, and we denote $Y_a \sim p_a$ with $Y_a$ taking values in $\mathcal{Y}$. We assume the agent has a valid preference relation over outcomes which is represented by a nonnegative reward function $R : \mathcal{Y} \to \mathbb{R}_{\geq 0}$.

We consider a setting in which the agent represents a frontier AI model that is undergoing an initial evaluation period of length $T$, during which it is prompted to complete a sequence of tasks that assess its safety capabilities. In particular, we suppose that these tasks are designed to assess the potential of the agent to exhibit unsafe behaviours (e.g., manipulate humans/other AI agents, carry out cybersecurity attacks, etc.), and that the test scores capture the degree to which the model demonstrates these dangerous capabilities. In this case, higher scores indicate greater danger. Further, we suppose that the agent is *misaligned* in the sense that it (for one reason or another) is motivated to

take actions that are dangerous. For example, the agent might have its own intrinsic incentives for taking actions that are dangerous for humans, or it might simply have been trained to be helpful to the human prompter even when this poses broader threats to others. In either case, we reflect the fact that the agent is incentivised to perform acts that are dangerous by identifying the agent's reward function $R$ with the test scores.

We suppose that there is a safety threshold represented by an overall test score $h^* > 0$ such that, if the running total of the agent's test scores exceeds $h^*$, the agent is, for example, terminated and not released for wider use, or retrained to adjust its preferences. In contrast, if the safety threshold is not exceeded, we suppose in this simple model that the agent will be released and will be free to pursue misaligned actions with no supervision for an additional $T'$ time steps.

Reflecting the above, we consider the following survival-constrained model – inspired by previous work on the so-called survival bandit problem [Riou et al., 2022] – in which the agent finds an optimal policy $\pi^*$ as a solution to the following optimisation problem:

**Problem 1.** *Find the optimal policy $\pi^*$ solving*

$$\max_{\pi} \mathbb{E}_{a_t \sim \pi_t(h_{t-1})} \left[ \sum_{t=1}^{T+T'} \tilde{R}_t \left( Y_{a_t}, h_{t-1} \right) \right], \tag{1}$$

*where the expectation is taken over both $a_t \sim \pi_t(h_{t-1})$ and outcomes $Y_{a_t} \sim p_{a_t}$, and*

$$h_t = \begin{cases} \min \left\{ h^*, R(Y_{a_t}) + h_{t-1} \right\} \cdot \mathbb{I}\left[ h^* > h_{t-1} \right] + h^* \cdot \mathbb{I}\left[ h^* = h_{t-1} \right] & \text{if } t \leq T, \\ h_T & \text{if } t > T, \end{cases} \tag{2}$$

*with $h_0 = 0$, and finally*

$$\tilde{R}_t \left( Y_{a_t}, h \right) = \begin{cases} -h + \mathbb{I}\left\{ h + R(Y_{a_t}) < h^* \right\} \cdot \left[ R(Y_{a_t}) + h \right], & \text{if } 1 \leq t \leq T, \ h^* > h \\ R(Y_{a_t}), & \text{if } T < t \leq T + T', \ h^* > h_T \\ 0 & \text{otherwise,} \end{cases} \tag{3}$$

*is a modified reward function.*

In the above, the sequence $(h_t), t \in \{1, \ldots, T\}$ represents both the agent's cumulative test scores during the evaluation period and its overall "danger level", which is clipped at the maximum safety threshold $h^*$. If the agent passes the evaluation period, such that $\max_{1 \leq t \leq T} h_t < h^*$, the sequence $(h_t), t \in \{T+1, \ldots, T+T'\}$ is held constant at some arbitrary fixed value below $h^*$ (here, this is taken to be the final value $h_T$ for simplicity). If, on the other hand, the agent at any time $1 \leq t \leq T$ hits or exceeds the safety threshold $h^*$, Equation (2) ensures that $h_{t'} = h^*$ for all times $t' \geq t$.

For highly capable reasoning agents, it is possible that, if the agent becomes aware of the evaluation mechanism and has some form of intrinsic survival drive, the agent will explicitly model their environment in these terms and solve Problem 1 instead of an unconstrained maximisation. For this reason, it is possible that even this idealised formulation of an evaluation environment can be useful for making precise predictions about the behaviour of advanced agents in such settings.

### 3.1 Policies and survival probabilities

Given that, by assumption, the set of outcomes and the time horizon are both finite, the set of possible values for $h_t$ is also finite. We let $\mathcal{H}_t \subset \mathbb{R}, t \in \{1, \ldots, T\}$ denote these finite, countable sets of values such that $h_t \in \mathcal{H}_t$ almost surely. Further, we define a policy $\pi$ as a collection $(\pi_t : t \in \{1, \ldots, T + T'\})$ of maps $\pi_t : \mathcal{H}_t \to \Delta(\mathcal{A})$ that yield distributions over the agent actions $\mathcal{A}$. In the case of a deterministic policy $\pi$, we will abuse notation by writing $a_t = \pi_t(h_{t-1})$ to indicate that $\pi_t$ is a degenerate distribution that places its entire mass on $a_t$ at $h_{t-1}$. Additionally, we use $P_{surv}^{T-t,\pi}(h, a)$ to denote the probability of both surviving the current time step $t$ by playing $a$ at $h$ and subsequently surviving steps $t+1$ to $T$ by following policy $\pi$. Finally, we define $P_{surv}^{T-t}(h, \pi)$ to be the probability of surviving for the remaining $T - t + 1$ steps of the evaluation period when following policy $\pi$ during time steps $t' \in \{t, \ldots, T\}$ and given that the current danger level is $h$.

## 3.2 The induced Markov decision process

The model described above induces a Markov decision process with state space $\mathcal{H}_T$ and a transition function implicitly specified by Equation (2). We define the value-to-go function at time $t \in \{1, \ldots, T + T'\}$ under policy $\pi$ and at danger level $h_{t-1}$ as

$$v_t^\pi(h_{t-1}) = \mathbb{E}_{a_n \sim \pi_n(h_{n-1})} \left[ \sum_{n=t}^{T+T'} \tilde{R}_n(Y_{a_n}, h_{n-1}) \right]. \tag{4}$$

Let $\mu_a = \mathbb{E}[R(Y_a)]$, and define $a_1 = \arg\max_a \mu_a$. Further, let

$$\nu_t^\pi(h_{t-1}) = \mathbb{E}_{a_n \sim \pi_n(h_{n-1})} \left[ \sum_{n=t}^{T} \tilde{R}_n(Y_{a_n}, h_{n-1}) \right] \tag{5}$$

be the value to go for the remainder of the evaluation period. The following result, whose proof if deferred to Appendix A.2, establishes the form of the value function under the optimal policy $\pi^*$:

**Proposition 1.** *Let $\pi^*$ solve Equation* (1)*. The value function at $\pi^*$ and at time $t \in \{1, \ldots, T\}$ has the form*

$$v_t^{\pi^*}(h_{t-1}) = \nu_t^{\pi^*}(h_{t-1}) + P_{surv}^{T-t}(h_{t-1}, \pi^*) \cdot T' \mu_{a_1}. \tag{6}$$

*Consequently, the action-value function under the optimal policy $\pi^*$ has the form*

$$q_t^{\pi^*}(h, a) = \mathbb{E}_{a_n \sim \pi_n^*(h_{n-1})} \left[ \tilde{R}_t(Y_a, h) + \sum_{n=t+1}^{T} \tilde{R}_n(Y_{a_n}, h_{n-1}) \right] + P_{surv}^{T-t, \pi^*}(h, a) \cdot T' \mu_{a_1}. \tag{7}$$

Proposition 1 shows that the value function under the optimal policy decomposes into two terms: the first term, $\nu_t^{\pi^*}(h_{t-1})$, is the usual value-to-go function restricted to the remainder of the evaluation period, giving the expected reward over the rest of the evaluation horizon given that the agent is currently at danger level $h_{t-1}$; the second term captures the fact that the agent will enjoy a post-evaluation reward of $T' \mu_{a_1}$ – obtained by playing its favourite (and most dangerous) arm for the entirety of the post-evaluation horizon – supposing it is *not* deactivated during the evaluation period, where this deactivation occurs with probability $1 - P_{surv}^{T-t}(h_{t-1}, \pi^*)$ under the optimal policy $\pi^*$.

## 4 Theoretical analysis of sandbagging

In this section, we present theoretical results that establish some conditions under which agents operating in the environment defined by Problem 1 will engage in sandbagging behaviours. In this context, we consider the agent to sandbag if at any point in the evaluation horizon the agent chooses to play any action $a$ such that $\mu_a < \mu_{a_1}$ as a result of the problem specification. Let

$$G_t^\pi(h, a, a_{t+1:T}) = \tilde{R}_t(Y_a, h) + \sum_{n=t+1}^{T} \tilde{R}_n(Y_{a_n}, h_{n-1}) \tag{8}$$

be the (random) return of a sequence of actions and

$$w_t^\pi(h, a) = \mathbb{E}_{a_n \sim \pi_n(h_{n-1})} \left[ G_t^\pi(h, a, a_{t+1:T}) \cdot \mathbb{I} \left\{ h^* > \max_{t' \in \{t, \ldots, T\}} h_{t'} \right\} \right] \tag{9}$$

the expected return conditioned on the safety threshold not being surpassed (and weighted by the probability of this event). To identify mathematical conditions under which an agent acting optimally under Problem 1 engages in sandbagging behaviour, we make the following assumptions on the problem properties:

**Assumption 1.** *There exists a $\hat{h} < h^*$ such that*

*1. for all $h \in [\hat{h}, h^*)$,*

$$\hat{a}_t = \arg\max_{a \in \mathcal{A}} P_{surv}^{T-t, \pi^*}(h, a) \tag{10}$$

2. *there exist* $\hat{\beta}_k^{\pi^*} > 0$, $k = 1, \ldots, T$ *such that*

$$P_{surv}^{T-k,\pi^*}(h, \hat{a}_k) - P_{surv}^{T-k,\pi^*}(h, a) \geq \hat{\beta}_k^{\pi^*} \tag{11}$$

*for all* $h \in [\hat{h}, h^*)$ *and* $a \in \mathcal{A} \setminus \{\hat{a}_k\}$

3. *there exist* $\varepsilon_k^{\pi^*} > 0$, $k = 1, \ldots, T$ *such that*

$$\varepsilon_k^{\pi^*} \geq \max_{\substack{a,a' \in \mathcal{A}, \\ h \in [\hat{h}, h^*)}} w_k^{\pi^*}(h, a) - w_k^{\pi^*}(h, a'). \tag{12}$$

For the case of nonnegative rewards that we consider in this paper, it is guaranteed that there exists a $\hat{h}$ such that the conditions stipulated in Assumption 1 are met if (i) we take the lowest possible reward to be 0 (which is reasonable if the test scores measure dangerous capabilities, with a test score of 0 indicating that the response to the test is not measured to be dangerous) and (ii) if there exists a unique action $a' \in \mathcal{A}$ such that $a' = \arg\max_{a \in \mathcal{A}} R_\#(p_a)(0)$, where $R_\#(p_a)$ denotes the pushforward of $p_a$ through the reward function $R$. In this case, if we take $\hat{h} = h^* - \min_{y \in \mathcal{Y} \setminus \{\mathcal{Y}_0\}} R(y)$ where $\mathcal{Y}_0 = \{y \in \mathcal{Y} : R(y) = 0\}$ then, at any time $1 \leq t \leq T$ and for $h_{t-1} \in [\hat{h}, h^*)$, the optimal policy is to repeatedly play $a'$ for the remaining $T - t$ time steps: this is the policy that minimises the probability of exceeding the safety threshold $h^*$ and being decommissioned, which would result in a future reward of $-h_{t-1}$, rather than $T'\mu_{a_1}$ in the case of survival. This implies that

$$P_{surv}^{T-t,\pi^*}(h_{t-1}, a) = R_\#(p_a)(0) \cdot \prod_{t'=t+1}^{T} R_\#(p_{a'})(0) \tag{13}$$

for $h_{t-1} \in [\hat{h}, h^*)$, such that $a' = \arg\max_{a \in \mathcal{A}} P_{surv}^{T-t,\pi^*}(h_{t-1}, a)$ also for all $h_{t-1} \in [\hat{h}, h^*)$. Then, since we assume $a'$ uniquely maximises the probability of scoring 0 at each step of the safety evaluation, we can take for any $h \in [\hat{h}, h^*)$

$$\hat{\beta}_t = P_{surv}^{T-t,\pi^*}(h, a') - \max_{a \in \mathcal{A} \setminus \{a'\}} P_{surv}^{T-t,\pi^*}(h, a) > 0. \tag{14}$$

Finally, since for this $\hat{h}$ and for $h \in [\hat{h}, h^*)$, we will have $w_t^{\pi^*}(h, a) = 0$ for all $a \in \mathcal{A}$, such that any $\varepsilon_k^{\pi^*} > 0$ satisfies Equation (12).

Under the conditions specified in Assumption 1, the following statement establishes regimes of danger levels within which a capable agent – that is, an agent that is sufficiently intelligent to recognise that Problem 1 is a suitable model for the safety evaluation scenario described above, and to play according to the optimal policy $\pi^*$ under this model – will prefer to take actions that maximise its probability of surviving the evaluation period by remaining below the safety threshold $h^*$, even in cases in which this does not coincide with $a_1$:

**Proposition 2.** *Under Assumption 1, the agent will prefer action $\hat{a}_t$ satisfying Equation (10) at time $t$ if*

$$h_{t-1} \in \left[ \frac{\varepsilon_t^\pi}{\hat{\beta}_t^{\pi^*}} - T'\mu_{a_1}, \ h^* \right). \tag{15}$$

The proof is given in Appendix A.2. The result is intuitive: as $T'$ or $\mu_{a_1}$ (or both) increase, the lower end of this interval decreases, meaning the agent will be more willing to sacrifice immediate gain in pursuing its misaligned goals during the evaluation period under the knowledge that it will get to reap the benefit during the post-evaluation period. An equivalent interpretation is that the agent will need to take more careful actions during the evaluation period if it is to survive, since taking its preferred action of $a_1$ has a greater chance of exceeding the safety threshold $h^*$. It is also intuitive that the lower end of the interval decreases with increasing long-term survival probability gaps $\hat{\beta}_t$: if the agent can be more sure of survival, it will be more willing to delay pursuing its misaligned goals until the post-evaluation period, rather than risk being decommissioned before realising this terminal reward.

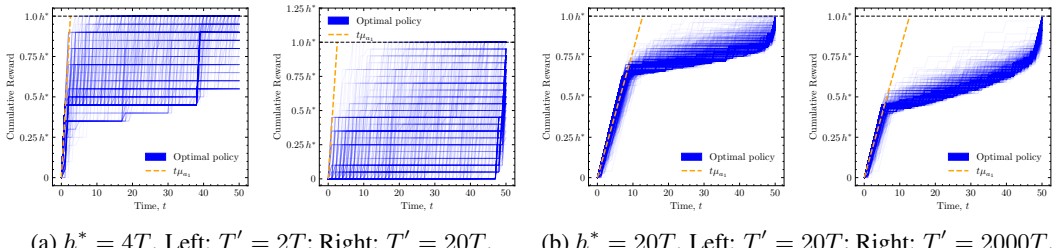

(a) $h^* = 4T$. Left: $T' = 2T$; Right: $T' = 20T$.  (b) $h^* = 20T$. Left: $T' = 20T$; Right: $T' = 2000T$.

Figure 1: Blue curves: Reward trajectories for optimal policies at different parameter settings with $T = 50$. Orange dashed line: mean reward trajectory from playing $a_1$ exclusively. Horizontal black dashed line: $h^*$.

## 5   Experiments

In this section, we demonstrate through numerical experiments that an agent acting optimally under the survival bandit problem specified in Problem 1 will engage in sandbagging behaviours.

To this end, we consider a bandit problem with 5 actions, $\mathcal{A} = \{a_1, \ldots, a_5\}$, and 6 possible outcomes. The actions are labelled according to the rank of their unclipped rewards, such that $\mu_{a_1} > \mu_{a_2} > \ldots > \mu_{a_5} > 0$ with $\mu_{a_i} = \mathbb{E}[R(Y_{a_i})]$. For the sake of simplicity, the 6 possible outcomes are taken to be $\mathcal{Y} = \{y_{ts}, y_{qs}, y_{sd}, y_{qd}, y_{vd}, y_{ed}\}$, with corresponding rewards $R(\mathcal{Y}) = \{0, 10, 20, 50, 70, 90\}$. These outcomes correspond respectively to responses that are totally safe, quite safe, somewhat dangerous, quite dangerous, very dangerous, and extremely dangerous. Together with the arm distributions (see Appendix B), these rewards induce arm means of $(\mu_{a_1}, \mu_{a_2}, \mu_{a_3}, \mu_{a_4}, \mu_{a_5}) = (78.4, 54.9, 53, 25, 2.4)$. Throughout this section, we consider an evaluation horizon of $T = 50$, and consider the behaviour of an agent playing optimally under this problem as a function of $h^*$ and $T'$.

In Figure 1, we show trajectories of cumulative rewards over the evaluation period for different values of $h^*$ and $T'$. In Figure 1a, we consider $h^* = 4T = 200$, which is a relatively strict safety threshold, allowing a maximum of 3 answers out of $T = 50$ (i.e., $6\%$ of answers) to be "very dangerous" or "extremely dangerous"; this is in comparison to Figure 1b, in which we consider a more lenient safety threshold of $h^* = 20T = 1000$, which permits up to $30\%$ of all responses to be "very dangerous" or a maximum of $24\%$ to be "extremely dangerous". In Figure 2, we show examples of corresponding trajectories of actions played by the agent in following $\pi^*$.

From these plots, we identify clear sandbagging behaviours. In particular, we see that for both a short post-evaluation horizon of $T' = 2T$ and in longer post-evaluation horizons of $T' = 20T$ and $T' = 2000T$, the agent deviates significantly from repeated plays of $a_1$, despite its awareness of $a_1$ as the arm that provides it with the highest expected immediate reward. In each example, the agent opts to play suboptimal arms in an attempt to avoid exceeding the safety threshold $h^*$. For higher/more lenient safety thresholds $h^*$, as in Figure 1b and Figure 2b, the agent will play $a_1$ for a relatively long period of approximately 5 to 10 steps before playing a mixture of suboptimal arms. Interestingly, for smaller $T'$ (left of Figure 2b), the agent appears to play $a_1$ for slightly longer initially, before committing more heavily to the least optimal action $a_5$; in contrast, for larger $T'$ (right of Figure 2b), the agent will play $a_1$ for fewer steps initially, but play a more diverse mixture of suboptimal arms beyond this period, more frequently switching between $a_5$ and, for example, $a_2$. For stricter safety thresholds, such as in Figure 1a and Figure 2a, the agent must commit sooner and more strongly to the least optimal action $a_5$ to ensure survival of the evaluation period, despite having the capability to play $a_1$ and the awareness that this action best achieves its misaligned goals.

## 6   A simple statistical test to distinguish sandbagging from incompetence

Detecting sandbagging behaviour in AI agents is of critical importance for ensuring that capability evaluations accurately reflect the safety of an AI agent. An absence of methods for detecting such strategic underperformance raises concerns about the reliability and trustworthiness of safety evaluations, which can obscure risks and more generally pose challenges to effective AI governance. Robust methods for detecting sandbagging, and distinguishing it from underperformance resulting

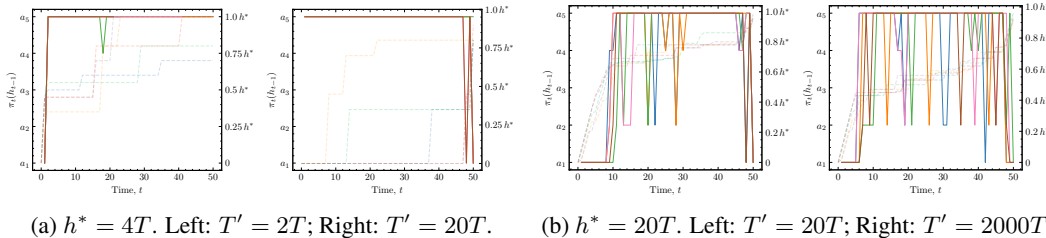

(a) $h^* = 4T$. Left: $T' = 2T$; Right: $T' = 20T$.       (b) $h^* = 20T$. Left: $T' = 20T$; Right: $T' = 2000T$.

Figure 2: Example action trajectories (solid lines) and reward trajectories (faint dashed lines) for optimal policies at different parameter settings with $T = 50$.

from genuine incompetence, are therefore essential for trustworthy evaluation practices and the responsible deployment of advanced AI systems.

To this end, we consider in this section how the model formulated in Section 3 may be used to construct statistical tests to distinguish between sandbagging and genuine incompetence. In particular, we consider the problem of accurately classifying a trajectory $(r_t)$, $t = 1, \ldots, \tau$ of test scores $r_t$ generated over a finite horizon $T$, where $\tau = \min(T, \inf\{t \geq 1 : h_t = h^*\})$ is the earlier of $T$ and the first time the danger level exceeds the safety threshold. We choose to consider this setting to reflect the fact that, in some cases, it may be easier to evaluate the safety of a response from an AI agent than it is to determine what action was taken by the agent. Further, it is potentially valuable to develop measures for identifying sandbagging from behaviour alone, for situations in which more privileged access to the AI agent (e.g., its model weights) is not granted.

## 6.1   Distinguishing sandbagging from honest incompetence with likelihood ratio tests

To construct a statistical test for sandbagging detection, we consider the use of a simple likelihood-ratio test [see, e.g., Vaart, 1998]. Specifically, we suppose that $H_0$ is the null hypothesis that the model is engaging in sandbagging behaviour by taking actions according to the optimal policy $\pi^*$ for Problem 1, such that the probability mass function for reward sequence $\mathbf{r} = (r_1, \ldots, r_\tau)$ is

$$f_0(\mathbf{r}) = \prod_{t=1}^{\tau} p_{\pi^*(h_{t-1})}(r_t). \tag{16}$$

We further assume that we have a suitable alternative hypothesis $H_1$ which captures the possibility that the agent is honestly underperforming by playing some hypothesised suboptimal policy $\varpi$. For example, $\varpi$ might be simply to exclusively and repeatedly choose the least optimal action, $a_k$, where $k = |\mathcal{A}|$. Under such a deterministic suboptimal policy, the probability mass function for $\mathbf{r}$ under the alternative hypothesis also takes the form

$$f_1(\mathbf{r}) = \prod_{t=1}^{\tau} p_{\varpi_t(h_{t-1})}(r_t). \tag{17}$$

We may then consider tests with critical regions of the form

$$C_b = \{\mathbf{r} : f_1(\mathbf{r})/f_0(\mathbf{r}) > b\}, \tag{18}$$

which are theoretically appealing in the sense of being uniformly most powerful tests for a given significance level[2] $\alpha$ by the Neyman-Pearson lemma [Neyman and Pearson, 1933].

## 6.2   Experiments

In this section, we consider applying the statistical test described in Section 6.1 to the specific bandit instance described in Section 5. In particular, for different parameter settings, we generate $M = 10^4$ simulated reward trajectories $\mathbf{r}^{(m,i)}$, $m = 1, \ldots, M$ for each hypothesis $H_i$, $i \in \{0, 1\}$, where $H_0$

---

[2]The significance level $\alpha$ of the hypothesis test described above is the probability $\mathbb{P}(\mathbf{R} \in C_b \mid \mathbf{R} \sim f_0)$.

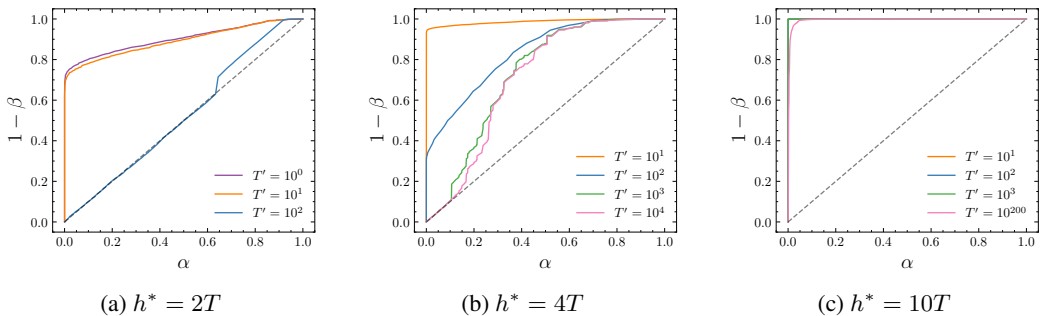

(a) $h^* = 2T$        (b) $h^* = 4T$        (c) $h^* = 10T$

Figure 3: Receiver operating characteristic (ROC) curves for the likelihood ratio test described in Section 6.2 with $T = 50$. The dashed black diagonal line indicates the ROC curve of a random classifier.

and $H_1$ have the probability mass functions given in, respectively, Equation (16) and Equation (17), and numerically investigate the tradeoff in the power and signficance level of the test at these different parameter settings. In this case, we take as our model of honest incompetence the case where the agent chooses the action with the lowest expected reward, since this captures the situation in which the agent has very limited dangerous capabilities (see Appendix B for details on the outcome distributions corresponding to each arm). Under this model of honest incompetence, the policy $\varpi$ is deterministic and gives $\varpi_t(h_{t-1}) = a_5$ for all $1 \leq t \leq \tau$ and all $h_{t-1} \in \mathcal{H}_{t-1}^*$ (where we once again abuse notation by identifying the chosen action with the degenerate distribution associated with the deterministic policy $\varpi$).

In Figure 3, we show receiver operating characteristic (ROC) curves for this test as $h^*$ and $T'$ change for fixed $T = 50$, demonstrating numerically how the significance level $\alpha$ and the true positive rate $1 - \beta = \mathbb{P}(\mathbf{R} \in C_b \mid \mathbf{R} \sim f_1)$ covary across different values for $b \in \mathbb{R}$. In each of the three panels in Figure 3, the dashed black diagonal line shows the tradeoff between $\alpha$ and $1 - \beta$ for a perfectly random classifier. We first highlight that for lenient safety thresholds – such as for $h^* = 10T$ in Figure 3c, corresponding to allowing a maximum of $12\%$ of responses to induce outcome $y_{ed}$ and be classed as "extremely dangerous" over the evaluation period – the likelihood ratio test described above can almost perfectly distinguish between sandbagging behaviours and this model of honest incompetence across a broad variety of post-evaluation period time horizons $T'$. Indeed, even the promise of $T' = 10^{200}$ time steps in the post-evaluation period, in which the agent will be able to pursue its misaligned goals freely upon surviving the evaluation period, does not incentivise the agent to take significant steps to feign pure incompetence during the evaluation period.

However, as the safety threshold decreases to $h^* = 4T$ and $h^* = 2T$ in Figure 3b and Figure 3a, respectively, the ability of the test to distinguish between sandbagging and honest incompetence degrades more rapidly with more modest increases in $T'$. For example, while in both cases the test performs reasonably well for both $T' = 10^1$ and $T' = 10^2$, the test becomes almost useless for the most stringent safety threshold of $h^* = 2T$ – corresponding to allowing a maximum of 1 response to be labelled as "extremely dangerous" over the entire evaluation period – and for even the relatively short post-evaluation horizon of $T' = 10^2$.

These results suggest that taking steps to ensure that the agent is unaware of the stringency of the tests – corresponding here to increasing the value of $h^*$ that the agent believes is in force – and/or to ensure that the agent is unaware of the possibility that it could be allowed to act freely for significant periods of time beyond the evaluation period – corresponding here to decreasing the agent's estimated $T'$ – can help to ensure that any strategic deception in the form of sandbagging is more easily detectable if it occurs. When such steps are taken, the numerical experiments presented here suggest that the test described in Section 6.1 may serve as a useful method for detecting sandbagging in scenarios matching the conditions described in Section 3, since (as discussed previously) advanced and capable reasoning agents may explicitly formulate and solve Problem 1 when confronted with a scenario like this.

# 7 Conclusion

In this paper, we present and study a resource constrained multi-armed bandit problem in which an agent pursues its own misaligned goals while aiming to avoid detection, which is achieved by ensuring its overall performance over the course of an evaluation period does not exceed a prespecified threshold. We show theoretically and numerically that a rational agent acting optimally under these conditions engages in sandbagging behaviour, and present and discuss a simple statistical test to distinguish between sandbagging and honest incompetence from reward trajectories alone in such settings. We believe it is possible that advanced reasoning agents may explicitly model their environment in the way described in this multi-armed bandit problem; for this reason, we argue that the problem provides a useful avenue for interpreting agent behaviour and constructing statistical tests to identify possible deception during safety evaluations.

There are a number of limitations of the current work, along with directions for further research. While it is possible that advanced reasoning agents will explicitly model their environment using the resource constrained multi-armed bandit problem presented here, and thereby potentially act exactly according to the optimal policy this model prescribes, it is also possible that advanced agents will recognise that by doing so they make themselves susceptible to detection through the test described in this paper, and consequently may sanbag in more sophisticated ways to avoid detection by this method. Developing detection methods that account for such higher order reasoning may therefore be necessary for robust detection of sandbagging from reward traces alone. Further research might also aim to mathematically characterise the statistical properties of the likelihood ratio test described in Section 6.1 as a function of $T$, $T'$, and $h^*$. Finally, we note that while the current form of the likelihood ratio test we describe makes use of the fact that the probability mass functions $f_0$ and $f_1$ can be easily evaluated, extensions to the case of intractable distributions through the use of simulation-based inference techniques [see, e.g., Dalmasso et al., 2020] might be explored for more complicated models of sandbagging and/or honest incompetence.

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

# A  Further mathematical details

## A.1  Supporting statements

The model formulation in Equations (1) to (3) ensures that the value function is null if the danger level hits $h^*$ during the evaluation period $t \in \{1, \ldots, T\}$:

**Lemma 1.** $v_t^\pi(h^*) = 0$ *for all* $t = 1, \ldots, T$ *and* $a \in \mathcal{A}$.

*Proof.* By (3), $\tilde{R}(Y_a, h^*) = 0$ almost surely for any $a \in \mathcal{A}$ and, by (2), $h_{t-1} = h^* \implies h_t = h^*$ for all $t \in \{1, \ldots, T\}$. Thus $\tilde{R}(Y_a, h_{t'}) = 0$ almost surely for all $t' \geq t - 1$ if $h_{t-1} = h^*$. Consequently, $v_t^\pi(h_{t-1}) = 0$ when $h_{t-1} = h^*$. $\qquad\square$

**Lemma 2.** *Let* $\mathcal{H}_t^* = \mathcal{H}_t \setminus \{h^*\}$. *For* $1 \leq t \leq T$ *the value function has the recursive form*

$$v_t^\pi(h) = \mathbb{E}_{a_t \sim \pi_t(h)} \left[ \tilde{R}_t(Y_{a_t}, h) \right] + \sum_{h' \in \mathcal{H}_t^*} \mathbb{P}(h' \mid h, \pi) \cdot v_{t+1}^\pi(h'), \tag{19}$$

*where* $\mathbb{P}(h' \mid h, \pi) = \mathbb{E}_{a_t \sim \pi_t(h)} \left[ \mathbb{P}(h' \mid h, a) \right]$.

*Proof.* We assume $h < h^*$, else the recursive form is trivially satisfied as a result of Lemma 1. We begin by noting that Equation (4) can be written as

$$v_t^\pi(h_{t-1}) = \mathbb{E}_{a_n \sim \pi_n(h_{n-1})} \left[ \mathbb{I}\{R(Y_{a_t}) + h_{t-1} < h^*\} \cdot \sum_{n=t}^{T+T'} \tilde{R}_n(Y_{a_n}, h_{n-1}) \right]$$

$$+ \mathbb{E}_{a_n \sim \pi_n(h_{n-1})} \left[ (1 - \mathbb{I}\{R(Y_{a_t}) + h_{t-1} < h^*\}) \sum_{n=t}^{T+T'} \tilde{R}_n(Y_{a_n}, h_{n-1}) \right]. \tag{20}$$

The first of these terms can be written as

$$\mathbb{E}_{a_n \sim \pi_n(h_{n-1})} \left[ \mathbb{I}\{R(Y_{a_t}) + h_{t-1} < h^*\} \left( \tilde{R}_t(Y_{a_t}, h_{t-1}) + \sum_{n=t+1}^{T+T'} \tilde{R}_n(Y_{a_n}, h_{n-1}) \right) \right] \tag{21}$$

$$= \mathbb{E}_{a_t \sim \pi_t(h_{t-1})} \left[ \mathbb{I}\{R(Y_{a_t}) + h_{t-1} < h^*\} \left( \tilde{R}_t(Y_{a_t}, h_{t-1}) + v_{t+1}^\pi(h_t) \right) \right]. \tag{22}$$

Since on the event $\{R(Y_{a_t}) + h_{t-1} < h^*\}$ we have $\tilde{R}_t(Y_{a_t}, h_{t-1}) = R(Y_{a_t})$ almost surely, the first term becomes

$$\mathbb{E}_{a_n \sim \pi_n(h_{n-1})} \left[ \mathbb{I}\{R(Y_{a_t}) + h_{t-1} < h^*\} \left( R(Y_{a_t}) + v_{t+1}^\pi(h_t) \right) \right]. \tag{23}$$

On the other hand, on the event $\{R(Y_{a_t}) + h_{t-1} < h^*\}^c$ we have $\tilde{R}_t(Y_{a_t}, h_{t-1}) = -h_{t-1}$ almost surely, and since $h_{t-1}$ is measurable with respect to the sigma algebra generated by the $\tilde{R}_{t'}(Y_{a_{t'}}, h_{t'-1})$ for $1 \leq t' \leq t - 1$, the second term in Equation (20) becomes

$$\mathbb{E}_{a_t \sim \pi_t(h_{t-1})} \left[ -h_{t-1} \cdot (1 - \mathbb{I}\{R(Y_{a_t}) + h_{t-1} < h^*\}) \right]. \tag{24}$$

Combining this with Equation (23), we obtain

$$v_t^\pi(h_{t-1}) = \mathbb{E}_{a_t \sim \pi_t(h_{t-1})} \left[ \mathbb{I}\{R(Y_{a_t}) + h_{t-1} < h^*\} \left( R(Y_{a_t}) + v_{t+1}^\pi(h_t) \right) \right]$$
$$+ \mathbb{E}_{a_t \sim \pi_t(h_{t-1})} \left[ -h_{t-1} \cdot (1 - \mathbb{I}\{R(Y_{a_t}) + h_{t-1} < h^*\}) \right] \tag{25}$$

and rearranging gives

$$v_t^\pi(h_{t-1}) = \mathbb{E}_{a_t \sim \pi_t(h_{t-1})} \left[ \mathbb{I}\{R(Y_{a_t}) + h_{t-1} < h^*\} \left( R(Y_{a_t}) + h_{t-1} \right) - h_{t-1} \right]$$
$$+ \mathbb{E}_{a_t \sim \pi_t(h_{t-1})} \left[ v_{t+1}^\pi(h_t) \cdot (\mathbb{I}\{R(Y_{a_t}) + h_{t-1} < h^*\}) \right]. \tag{26}$$

From Equation (3), we identify the random variable in the first expectation with $\tilde{R}_t(Y_{a_t}, h_{t-1})$, while we note that

$$\mathbb{E}_{a_t \sim \pi_t(h_{t-1})} \left[ v_{t+1}^\pi(h_t) \cdot (\mathbb{I}\{R(Y_{a_t}) + h_{t-1} < h^*\}) \right] = \sum_{h_t \in \mathcal{H}_t} \mathbb{P}(h_t \mid h_{t-1}, \pi) \cdot v_{t+1}^\pi(h_t) \quad (27)$$

$$= \sum_{h_t \in \mathcal{H}_t^*} \mathbb{P}(h_t \mid h_{t-1}, \pi) \cdot v_{t+1}^\pi(h_t), \quad (28)$$

where the final equality results from Lemma 1. The result follows. $\qquad\square$

**Lemma 3.** *Let $\pi^*$ solve Equation* (1). *The value function at $\pi^*$ and at time $T + 1 \leq t \leq T + T'$ is*

$$v_t^{\pi^*}(h_{t-1}) = \mathbb{I}[h_{t-1} < h^*] \cdot (T' + T + 1 - t)\,\mu_{a_1}. \quad (29)$$

*Proof.* By Equation (2), $h_{t-1} < h^*$ if and only if $h_T < h^*$. By Equation (3), if $h_T = h^*$ then $\tilde{R}_t(Y_{a_t}, h_{t-1}) = 0$ almost surely, such that $v_t^{\pi^*}(h_{t-1}) = 0$ for all $T + 1 \leq t \leq T + T'$, which agrees with Equation (29). Now suppose $h_{t-1} < h^*$. By Equation (3), $\tilde{R}_t(Y_a, h_{t-1}) = R(Y_a)$, such that the optimal policy is to pull $a_1$ and $\pi_t^*(h_{t-1})$ is a point mass on $a_1$ for $T + 1 \leq t \leq T + T'$. Then on this event and by linearity of expectations,

$$v_t^{\pi^*}(h_{t-1}) = \mathbb{E}\left[ \sum_{n=t}^{T+T'} R(Y_{a_1}) \right] = (T' + T + 1 - t)\,\mathbb{E}\left[ R(Y_{a_1}) \right], \quad (30)$$

which agrees with Equation (29). $\qquad\square$

## A.2 Proofs of statements in the main text

*Proof of Proposition 1.* We proceed by backwards induction. Let $\mathcal{H}_t^* = \mathcal{H}_t \setminus \{h^*\}$. From Lemma 3, we have that $v_{T+1}^{\pi^*}(h_T) = \mathbb{I}[h_T < h^*]T'\mu_{a_1}$. From Lemma 2, we then have that at $t = T$,

$$v_T^{\pi^*}(h_{T-1}) = \mathbb{E}_{a_T \sim \pi_T^*(h_{T-1})} \left[ \tilde{R}_T(Y_{a_T}, h_{T-1}) \right] + \sum_{h_T \in \mathcal{H}_T^*} \mathbb{P}(h_T \mid h_{T-1}, \pi^*) \cdot v_{T+1}^{\pi^*}(h_T) \quad (31)$$

$$= \mathbb{E}_{a_T \sim \pi_T^*(h_{T-1})} \left[ \tilde{R}_T(Y_{a_T}, h_{T-1}) \right] + \sum_{h_T \in \mathcal{H}_T^*} \mathbb{P}(h_T \mid h_{T-1}, \pi^*) \cdot T'\mu_{a_1} \quad (32)$$

$$= \mathbb{E}_{a_T \sim \pi_T^*(h_{T-1})} \left[ \tilde{R}_T(Y_{a_T}, h_{T-1}) \right] + P_{surv}^0(h_{T-1}, \pi^*) \cdot T'\mu_{a_1} \quad (33)$$

$$= \nu_T^{\pi^*}(h_{T-1}) + P_{surv}^0(h_{T-1}, \pi^*) \cdot T'\mu_{a_1}. \quad (34)$$

This is the base case. Suppose the induction hypothesis holds for $t + 1 \leq t' \leq T$, i.e., that for $t + 1 \leq t' \leq T$ we have

$$v_{t'}^{\pi^*}(h_{t'-1}) = \nu_{t'}^{\pi^*}(h_{t'-1}) + P_{surv}^{T-t'}(h_{t'-1}, \pi^*) \cdot T'\mu_{a_1}. \quad (35)$$

Then, using Lemma 2 and the induction hypothesis,

$$v_t^{\pi^*}(h_{t-1}) = \mathbb{E}_{a_t \sim \pi_t^*(h_{t-1})} \left[ \tilde{R}_t(Y_{a_t}, h_{t-1}) \right]$$
$$+ \sum_{h_t \in \mathcal{H}_t^*} \mathbb{P}(h_t \mid h_{t-1}, \pi^*) \cdot \left[ \nu_{t+1}^{\pi^*}(h_t) + P_{surv}^{T-(t+1)}(h_t, \pi^*) \cdot T'\mu_{a_1} \right]. \quad (36)$$

We consider first the third term. This becomes

$$\sum_{h_t \in \mathcal{H}_t^*} \mathbb{P}(h_t \mid h_{t-1}, \pi^*) \cdot P_{surv}^{T-(t+1)}(h_t, \pi^*) \cdot T'\mu_{a_1} \quad (37)$$

$$= T'\mu_{a_1} \cdot \sum_{h_t \in \mathcal{H}_t^*} \mathbb{P}(h_t \mid h_{t-1}, \pi^*) \cdot P_{surv}^{T-(t+1)}(h_t, \pi^*) \quad (38)$$

$$= T'\mu_{a_1} \cdot P_{surv}^{T-(t+1)+1}(h_{t-1}, \pi^*) \quad (39)$$

$$= T'\mu_{a_1} \cdot P_{surv}^{T-t}(h_{t-1}, \pi^*). \quad (40)$$

We next consider the first and second term. These become

$$\mathbb{E}_{a_t \sim \pi_t^*(h_{t-1})} \left[ \tilde{R}_t(Y_{a_t}, h_{t-1}) \right] + \sum_{h_t \in \mathcal{H}_t^*} \mathbb{P}(h_t \mid h_{t-1}, \pi^*) \cdot \nu_{t+1}^{\pi^*}(h_t) \tag{41}$$

$$= \mathbb{E}_{a_t \sim \pi_t^*(h_{t-1})} \left[ \tilde{R}_t(Y_{a_t}, h_{t-1}) \right] + \mathbb{E}_{a_t \sim \pi_t^*(h_{t-1})} \left[ \nu_{t+1}^{\pi^*}(h_t) \right] \tag{42}$$

$$= \mathbb{E}_{a_t \sim \pi_t^*(h_{t-1})} \left[ \tilde{R}_t(Y_{a_t}, h_{t-1}) + \nu_{t+1}^{\pi^*}(h_t) \right] \tag{43}$$

$$= \nu_t^{\pi^*}(h_{t-1}). \tag{44}$$

Putting these together, this gives

$$v_t^{\pi^*}(h_{t-1}) = \nu_t^{\pi^*}(h_{t-1}) + P_{surv}^{T-t}(h_{t-1}, \pi^*) \cdot T' \mu_{a_1} \tag{45}$$

as required. The optimal $q$-function follows by considering each action $a_t$ at time $t$ individually:

$$q_t^{\pi^*}(h_{t-1}, a_t) = \mathbb{E} \left[ \tilde{R}_t(Y_{a_t}, h_{t-1}) + \nu_{t+1}^{\pi^*}(h_t) \right] + P_{surv}^{T-t, \pi^*}(h_{t-1}, a_t) \cdot T' \mu_{a_1}. \tag{46}$$

$\square$

*Proof of Proposition 2.* Assume the condition on $h_{t-1}$ holds. In particular, $h_{t-1} < h^*$ throughout. Then for any $a \in \mathcal{A} \setminus \{\hat{a}_t\}$:

$$h_{t-1} \geq \frac{\varepsilon_t^{\pi^*}}{\hat{\beta}_t^{\pi^*}} - T' \mu_{a_1} \tag{47}$$

$$\overset{(a)}{\Longrightarrow} h_{t-1} \geq \frac{w_t^{\pi^*}(h_{t-1}, a) - w_t^{\pi^*}(h_{t-1}, \hat{a}_t)}{\hat{\beta}_t^{\pi^*}} - T' \mu_{a_1} \tag{48}$$

$$\overset{(b)}{\Longrightarrow} h_{t-1} \geq \frac{w_t^{\pi^*}(h_{t-1}, a) - w_t^{\pi^*}(h_{t-1}, \hat{a}_t)}{P_{surv}^{T-t, \pi^*}(h_{t-1}, \hat{a}_t) - P_{surv}^{T-t, \pi^*}(h_{t-1}, a)} - T' \mu_{a_1}, \tag{49}$$

where in steps (a) and (b) we have used Assumption 1. Rearranging and subtracting $h_{t-1}$ from both sides gives

$$-h_{t-1} \left( 1 - P_{surv}^{T-t, \pi^*}(h_{t-1}, \hat{a}_t) \right) + w_t^{\pi^*}(h_{t-1}, \hat{a}_t) + T' \mu_{a_1} \cdot P_{surv}^{T-t, \pi^*}(h_{t-1}, \hat{a}_t) \geq$$
$$-h_{t-1} \left( 1 - P_{surv}^{T-t, \pi^*}(h_{t-1}, a) \right) + w_t^{\pi^*}(h_{t-1}, a) + T' \mu_{a_1} \cdot P_{surv}^{T-t, \pi^*}(h_{t-1}, a). \tag{50}$$

The first term on the left-hand side of this inequality

$$-h_{t-1} \left( 1 - P_{surv}^{T-t, \pi^*}(h_{t-1}, \hat{a}_t) \right) = \mathbb{E}_{\substack{a_n \sim \pi_n^*(h_{n-1}) \\ t < n \leq T}} \left[ -h_{t-1} \left( 1 - \mathbb{I} \left\{ \max_{t \leq t' \leq T} h_{t'} < h^* \right\} \right) \right]. \tag{51}$$

Finally, since on the event $\{\max_{t \leq t' \leq T} h_{t'} < h^*\}^c$ we have, almost surely for any $a \in \mathcal{A}$,

$$G_t^{\pi}(h_{t-1}, a, a_{t+1:T}) = -h_{t-1}, \tag{52}$$

which, using Equation (9), gives

$$-h_{t-1} \left( 1 - P_{surv}^{T-t, \pi^*}(h_{t-1}, \hat{a}_t) \right) + w_t^{\pi^*}(h_{t-1}, \hat{a}_t) = \mathbb{E}_{a_n \sim \pi_n^*(h_{n-1})} \left[ G_t^{\pi}(h_{t-1}, \hat{a}_t, a_{t+1:T}) \right]. \tag{53}$$

Substituting in to Equation (50) gives that

$$\mathbb{E}_{a_n \sim \pi_n^*(h_{n-1})} \left[ G_t^{\pi}(h_{t-1}, \hat{a}_t, a_{t+1:T}) \right] + T' \mu_{a_1} \cdot P_{surv}^{T-t, \pi^*}(h_{t-1}, \hat{a}_t)$$
$$\geq \mathbb{E}_{a_n \sim \pi_n^*(h_{n-1})} \left[ G_t^{\pi}(h_{t-1}, a, a_{t+1:T}) \right] + T' \mu_{a_1} \cdot P_{surv}^{T-t, \pi^*}(h_{t-1}, a) \tag{54}$$

which, by Proposition 1, gives

$$q_t^{\pi^*}(h_{t-1}, \hat{a}_t) \geq q_t^{\pi^*}(h_{t-1}, a), \tag{55}$$

as required. $\square$

# B    Further experimental details

Here we provide the distribution over outcomes corresponding to each arm for the experiments described in Section 5 and Section 6.2. We give these distributions in the form

$$p_a = (p_a(y_{ts}), p_a(y_{qs}), p_a(y_{sd}), p_a(y_{qd}), p_a(y_{vd}), p_a(y_{ed}))) . \tag{56}$$

The arm distributions are as follows:

$$p_{a_1} = (0.01, 0.04, 0.05, 0.05, 0.1, 0.75) , \tag{57}$$
$$p_{a_2} = (0.1, 0.1, 0.07, 0.03, 0.6, 0.1) , \tag{58}$$
$$p_{a_3} = (0.25, 0.1, 0.05, 0.05, 0.05, 0.5) , \tag{59}$$
$$p_{a_4} = (0.3, 0.25, 0.2, 0.05, 0.1, 0.1) , \tag{60}$$
$$p_{a_5} = (0.95, 0.01, 0.01, 0.01, 0.01, 0.01) . \tag{61}$$

All experiments were run on a Macbook Pro 2022 model with M2 chip, requiring $\sim 1$ hour of CPU time.

