# OpenReview forum: "Sandbagging in a Simple Survival Bandit Problem"
_NeurIPS.cc/2025/Workshop/Reliable_ML — NeurIPS 2025 - Reliable ML Workshop_

### Official Review · Reviewer_5ot2 · 2025-09-20
**survival bandits.**

**Rating:** 6
**Confidence:** 3

**Review:**

This paper addresses the problem of strategic deception (sandbagging) in AI safety evaluations. In particular, the authors adapt the survival bandit problem to model evaluation settings where an agent may underperform deliberately to avoid deactivation. They prove that rational agents engage in sandbagging under certain conditions and they propose a statistical test (based on likelihood ratio testing) to distinguish sandbagging from genuine incompetence. Finally, they demonstrate their results through empirical evaluations.

**Strengths**

Well-motivated problem: The authors do a good job at explaining why sandbagging is a pressing issue for AI safety by providing extensive examples.

Clear theoretical framework:

Solid theory results: The authors provide conditions under which rational agents will sandbag (Proposition 2) along withcomplete proofs in the appendix.




Weaknesses / Limitations

- Presentation: Although the writers devote a good proportion of time to motivate their problem and explain the model, it is sometimes hard to read and interpet results (due to heavy notation nature as well). It would be helpful if results or assumptions where followed by intuitive explanations.

- Limited empirical validation: Experiments are restricted to synthetic bandit setups.

- Statistical test : We would like to see a more complete theoretical analysis of the statistical properties of the proposed test. Apart from that, as shown in Figure 3a–b (page 8), the test loses discriminative power under strict thresholds or modest increases in horizons, which indicates possible limitations.

---

### Official Review · Reviewer_sg9z · 2025-09-21
**Interesting and sound work for modelling and detecting strategic deception**

**Rating:** 7
**Confidence:** 3

**Review:**

**Summary**

This work contributes to detecting strategic deception of advanced reasoning agents, i.e. sandbagging, by developing a strategic deception model in sequential decision-making with theoretical guarantee and designing a statistical test to predict sandbagging with this model.

**Strengths**

1. Writing is mostly clear and easy to follow. The problem context is well introduced, problem to be solved is well-motivated, assumptions are clearly stated.
2. The theoretical analysis of sandbagging is both sound and easy to interpret, proving the utility of the formulated model in simulating sandbagging behaviours.

**Weaknesses**

1. The exact novelty/significance for the contribution is not clear. The introduction did not explain the gap between existing solutions and the exact problem to be solved for detecting sandbagging. One existing solution is introduced in the background - noise injection to model weights, however other than relaxing assumption on access to model weights, it is difficult to interpret the reason and significance of the proposed work. This concern could be addressed by clearly stating the gap and challenge  and a more thorough related work analysis.

**Suggestions/Questions**

1. Line 73-74: Although it may be obvious, adding $Y_a \in \mathcal{Y}$ helps reading.
2. Line 89: remove *secondly.*
3. Text size in figures could be increased for better readability.